# Automaticity and Executive Abilities in Developmental Dyslexia: A Theoretical Review

**DOI:** 10.3390/brainsci12040446

**Published:** 2022-03-27

**Authors:** James H. Smith-Spark, Rebecca Gordon

**Affiliations:** 1Division of Psychology, School of Applied Sciences, London South Bank University, London SE1 0AA, UK; 2Department of Psychology and Human Development, Institute of Education, University College London, London WC1H 0AA, UK; rebecca.gordon@ucl.ac.uk

**Keywords:** developmental dyslexia, executive function, Model of the Control of Action, Supervisory Attentional System, working memory, prospective memory, strategy use, prefrontal cortex, cerebellum

## Abstract

Cognitive difficulties are well documented in developmental dyslexia but they present a challenge to dyslexia theory. In this paper, the Model of the Control of Action is proposed as a theoretical explanation of how and why deficits in both automaticity and executive abilities are apparent in the cognitive profiles of dyslexia and how these deficits might relate to literacy difficulties. This theoretical perspective is used to consider evidence from different cognitive domains. The neuroanatomical underpinnings of automaticity and executive abilities are then discussed in relation to the understanding of dyslexia. Links between reading, writing, and executive function are considered. The reviewed evidence suggests that dyslexia theory should consider an interaction between procedural learned behaviour (automaticity) and higher-order (executive) abilities. The capacity to handle environmental interference, develop and engage adaptive strategies accordingly, and plan actions all require interactions between the cerebellum and the prefrontal cortex (PFC). Difficulties in these areas might explain both impairments in the cumulative development of literacy skills in childhood and general task management in everyday life in adulthood. It is suggested that improved measures are required to assess this cerebellar–PFC interaction and to allow early identification of future literacy difficulties, allowing implementation of timely interventions and reasonable adjustments.

## 1. Introduction

Developmental dyslexia is a frequently occurring neurodevelopmental condition which is characterised by literacy-related problems that cannot be accounted for in terms of low intelligence, socioeconomic status or emotional problems (e.g., [1,2]). The purpose of this review is to examine the evidence from explanations that go beyond the traditional hypotheses based on phonology. For example, the formulation of the phonological deficit hypothesis argues for core problems in dyslexia with phonological processing ([3]; for reviews, see [4,5]). Beyond the literacy problems that are central to dyslexia, children and adults with the condition have been found to show a range of broader cognitive difficulties affecting, for example, memory (e.g., [6,7,8]), attention (e.g., [9]), and executive function (e.g., [10,11,12]). There are several theories of dyslexia which can explain, to varying extents, these broader cognitive impairments (e.g., [13,14,15,16]). The current paper focuses on the evidence in support of a hypothesis originally proposed by Smith-Spark and Fisk ([17]; see also [18]), which argues for a dyslexia-related deficit in a component of the Model of the Control of Action [19], namely the Supervisory Attentional System (SAS). According to this argument, deficits in the SAS might explain impairments in abilities that are executive in nature, as the SAS is called into play when poorly learned or novel actions are necessary. The current paper explores this proposition further, by including evidence for other important dyslexia-related deficits, including automaticity (e.g., [14]), and considering how these deficits might be explained by the complete Model of the Control of Action, rather than focusing on just the SAS component. The effects of dyslexia are lifelong and can present different challenges to cognition than those experienced in childhood (e.g., [20]). As a consequence, the purpose of this theoretical review is to explain more fully the cognitive profile of dyslexia by examining evidence from both child and adult populations. The paper begins by providing an explanation of the Model of the Control of Action. Evidence of dyslexia-related impairments, drawn from a range of cognitive constructs, is then considered specifically in the light of SAS dysfunction. To complement this review of the behavioural literature, the neuroanatomy of executive function, and specifically the pre-frontal cortex (PFC), is then discussed and links are made to the neuroanatomy of dyslexia. To understand the evidence for dyslexia-related deficits in automaticity, evidence for atypical neuroanatomy of the cerebellum is examined, along with its role in adaptive behaviours. The paper concludes by discussing how these deficits inter-relate to explain broader observed deficits and how they might link to the impairments in literacy-related processes which lie at the heart of dyslexia.

## 2. The Model of the Control of Action

Norman and Shallice’s [19] Model of the Control of Action argues that two complementary processes cooperate to control action, namely contention scheduling and the SAS. Contention scheduling is the action selection mechanism which permits the control of relatively simple, well-learned actions. The SAS interrupts local level control over cognition (i.e., that performed by contention scheduling) whenever environmental conditions make the cognitive system vulnerable to distraction or stereotyped behaviour, and adaptive behaviours are required to meet task demands.

Contention scheduling is, then, the system responsible for executing learned behaviours. It has been argued that contention scheduling can account not only for the action sequencing of daily routines but also for the errors in behaviour which are observed in people with cognitive impairments [21]. Contention scheduling is conceptualised as consisting of a hierarchy of schemata from which the appropriate action is selected based on environmental triggers. The schemata themselves are made up of components that may overlap with other schemata. For example, the schema for unscrewing the top of a bottle of household cleaner and the schema for unscrewing the top of a bottle of water would share the holding and unscrewing components; however, it would not be desirable for these two actions to share a component for drinking. The necessary change in action occurs when the triggering of a component exceeds the threshold of a schema, causing it to excite and activate the component of another. A schema continues to be activated until either its objective is achieved or the objective is taken on by another component. This flow of actions, therefore, requires the concurrent activation and inhibition of related components in order to achieve the ultimate goal (the neuroanatomical underpinnings of the process are described in Section 5.1). If these overlapping processes do not work in harmony, then conflicting schemata may become activated simultaneously and, as a result, disrupt the ultimate aim of the actions. Likewise, similar disruption ensues if the environmental triggers do not activate schemata at the specific time necessary to carry out the appropriate action. Such disruption can occur when the surrounding environment demands a set of actions for which there are currently no schemata. When such non-routine situations occur, the creation of temporary schemata is required. Within the Model of the Control of Action, the system responsible for generating such schemata is the SAS [21,22,23]. In order to create these new schemata, it is argued that the SAS controls, coordinates, and integrates information and regulates and directs attention in order to control and promote behaviour consistent with known goals [19]. In order to achieve its objectives, it has been argued [19] that the SAS is involved in goal-directed behaviours, such as the inhibition of habitual, inappropriate responses and error correction, which in turn enable higher-order abilities such as prioritisation, decision making and planning.

Evidence in support of the existence of the SAS comes from neuroimaging and neuropsychological studies that have attempted to identify cortical regions involved in these high-level, goal-directed tasks. This evidence arises from a meta-analysis of neuroimaging studies [24] that found that specific regions of the frontal lobes were activated during tasks requiring the suppression of routine behaviours and the implementation of non-routine action. For example, the Stroop task [25] and stop-signal tasks [26] are both commonly used as measures of inhibitory control as they are designed to draw on the ability to countermand a pre-potent or routine response in favour of another, more task-oriented response [27]. This ability has been related to activity in the PFC during stop-signal tasks [28] and the Stroop task [29]. The findings of Norman and Shallice [19] are consistent with neuropsychological case studies which have found that patients with frontal lobe damage perform poorly on such tasks compared with healthy controls (see [30] for a review).

Although the Model of the Control of Action [19] explains performance in many executive tasks, it should, however, be noted that there is some difficulty in translating these theoretical explanations into laboratory-based measures. This is because rigid laboratory conditions differ from those presented in real-world environments and this difference is likely to be reflected in the cognitive resources required in the laboratory compared with real life (see [31] for a review). This is especially so in populations where cognitive deficits are evident (e.g., [23,30]). As a consequence, there is much value to be derived from the use of ecologically valid measures which may better reflect the control of attention required in everyday life, by tapping abilities related to multiple cognitive constructs. This is discussed in the context of a bidirectional relationship between automaticity and cortical control in Section 5.

## 3. SAS Dysfunction in Dyslexia

Smith-Spark and Fisk [17] argued for dyslexia-related SAS dysfunction based on a difference in performance between adults with and without dyslexia on a spatial working memory measure. In this task, the participants were required to indicate whether more cells were highlighted above or below a central dividing line displayed on a computer screen, while also being asked to remember the location of a differently highlighted cell for later serial recall. Despite no overall group difference in scores being found, an interaction was uncovered between group membership and performance in the first or second half of the test. The group without dyslexia showed no difference in their recall accuracy between the first and second halves of the task. The recall of the group with dyslexia, on the other hand, was lower than that of the group without dyslexia on the first half of the task but improved to a level of performance equivalent to that of the group without dyslexia in the second half. To explain this pattern of performance, Smith-Spark and Fisk argued that the task demands were very arbitrary and quite unlike the kind of task demands which people are likely to face in their day-to-day lives. Given this task novelty, the authors proposed that none of their participants had existing strategies in place to facilitate performance and, instead, had to generate a new schema to carry out the task. Smith-Spark and Fisk argued that a dyslexia-related deficit in the SAS could explain their findings, with the participants with dyslexia being slower than the participants without dyslexia to instantiate a new schema in response to these novel task demands. Coping with task novelty is posited as a defining characteristic of executive function [23,32,33,34]. Executive function, in turn, is argued to be representative of the high-level cognitive behaviours governed by the SAS within the Model of the Control of Action [35,36]. Executive function and dyslexia is addressed in detail in Section 4.2.

A dyslexia-related impairment in SAS function has been raised independently in a study with children with dyslexia. Varvara et al. [18] (p. 5) proposed the possibility that “a more global deficit in higher-order cognitive mechanisms” might be a critical feature of dyslexia, as an alternative explanation to central executive problems accounting for working memory difficulties. They identified impairments in the executive and attentional control functions of the SAS as a candidate for dyslexia-related reading difficulties.

## 4. Further Evidence

Further evidence that could be taken as supporting the argument for SAS dysfunction in dyslexia is presented in this section, drawn from a range of cognitive domains. Initially, impairments in working memory are discussed as these include early evidence for phonological deficits. However, this evidence suggests broader cognitive impairments, which leads to considerations of problems related to executive function. Evidence for difficulties in abilities that draw on executive function, specifically strategy selection, metacognition, and prospective memory, is then reviewed.

### 4.1. Working Memory

The multicomponent model of working memory [35,37,38] is the most enduring account of this construct and has an evidence base that spans almost five decades [39]. It is of benefit briefly to describe that model [37], and subsequent revised versions [35,38], in order to provide the context for considering studies that have identified deficits in this ability in people with dyslexia. The multicomponent model consists of three elements: an attentional control system, referred to as the central executive, and two sub-systems responsible for the temporary storage of phonological and visuospatial material. These latter two systems are known as the phonological loop and the visuo-spatial sketchpad, respectively. The central executive is defined as directing attention to relevant information, suppressing irrelevant information, inhibiting reaction to irrelevant information, and switching attention between different processes [38,40]. Inhibition, in this instance, refers to the deliberate restraint of prepotent and/or automatic responses in order to attend to task-relevant stimuli [41,42]. Task-switching describes the ability to alternate between cognitive processes in order to apply an appropriate action to a certain situation [43,44]. It is important to note that the role of the SAS in regulating and directing attention (e.g., [23,45]; as described in Section 2) has been equated by Baddeley [35,36] with the central executive in the multicomponent model of working memory [34].

Working memory difficulties are well documented in dyslexia, in children and adolescents ([46,47]; see [6] for a review) and in adults [e.g., [17,48]. Typically, dyslexia-related impairments in working memory have been interpreted as indicating a core phonological deficit that can be identified via the administration of verbal short-term memory tasks [e.g., [8,49,50,51,52,53,54,55]. However, there is also some evidence for visuospatial working memory deficits in this population. For example, variance in reading ability has been explained by performance on measures of visuospatial attention in children with developmental dyslexia [7,18,56]. These problems have been found to persist into adulthood, with lower levels of performance evident on complex figure tests [57,58] and spatial working memory span tasks [8,17]. For example, dyslexia-related phonological working memory problems have been reported across a battery of tasks administered to age- and IQ-matched adults with and without dyslexia [8]. The performance of the adults with dyslexia was also found to be lower on a visuospatial working memory task when the processing and memory loads were most demanding, suggesting a central executive component to the working memory difficulties experienced by people with dyslexia. Similar findings have been reported in a study [59] which assessed adults with developmental dyslexia on a range of verbal and visuospatial tasks. Along with showing the expected phonological deficits, the participants with dyslexia also demonstrated particular difficulty with performing visuospatial working memory and novel visual patterns tasks. As these cognitive abilities rely on attentional control [60], this again suggests a broader executive deficit. Indeed, Varvara et al. have argued [18] that a failure of the central executive to supervise the functioning of the phonological loop and the visuospatial sketchpad provided a full explanation of the executive function problems manifested by the children with dyslexia in their study. Furthermore, it has been found that, compared with typically developing children, children with reading difficulties showed domain-general working memory deficits (i.e., verbal and visuospatial) and performed less well on a task designed to measure central executive function [61].

Domain-general working memory deficits have, thus, been found in dyslexia, with problems documented across both the phonological and visuospatial processing domains. As noted previously, working memory, and specifically the central executive (cf. [35,36]), can be viewed as one aspect of a broader conceptualisation of higher-order cognition, namely executive function (e.g., [62,63,64]), and such abilities are considered in Section 4.2.

### 4.2. Executive Function

Although there is no ubiquitous agreed definition of executive function, it can be described broadly as a set of higher-order cognitive control processes that enable purposeful behaviours, required to achieve a known goal [64,65,66]. As stated in Section 4.1., Baddeley has equated the central executive to executive function [35,36,67,68] and the SAS [34]. Furthermore, the SAS is commonly considered as being directly analogous to executive function (e.g., [69]). The high degree of congruence in the way that the central executive and executive function are defined is extensively acknowledged and, though tending to be studied in separate literatures, their analogous nature is widely accepted [69]. In addition, there are many behavioural, neuropsychological, neurobiological, neuroimaging and computer modelling studies to support the argument that cognitive control, whether defined as the central executive, executive function or the SAS, is an intrinsic requirement for optimal functioning in everyday life [70,71,72] (see [73] for a review).

Executive function problems are well documented in children with dyslexia (for reviews, see [10,74]). More generally, there is a small body of laboratory-based research reporting that children with dyslexia have problems with the higher-order cognitive ability of planning [75,76,77]. Furthermore, difficulties with executive function have also been found to affect children with dyslexia in everyday life. The Behavior Rating Inventory of Executive Function (BRIEF; [78]) has been administered [79] to parents of either typically developing children, children with poor word recognition abilities or children with specific reading comprehension difficulties but with average word recognition abilities. The parents were asked to rate the frequency with which a range of everyday executive function problems were shown by their offspring, who were aged 10 to 14 years. The parents of children with poor word recognition skills rated their children as experiencing more frequent executive function problems overall than parents of either children with reading comprehension problems or typically developing children. More frequent executive function problems have also been reported on the BRIEF by parents and teachers of children with dyslexia compared with those of children without dyslexia [80].

Executive function problems have also been reported in adults with dyslexia but these problems have tended to vary by task (e.g., [11,81]). However, a broad range of executive function deficits in the same sample of adults with dyslexia have been found [12]. Compared with IQ-matched adults without dyslexia, adults with dyslexia showed poorer inhibition, updating, and set shifting abilities on laboratory-based tests. The deficits found in this study across these abilities will now be considered in relation to the SAS. With regard to inhibition, the participants with dyslexia were significantly less accurate in inhibiting a prepotent response in favour of making an alternative motor response during the inhibition phase of a variation on a Go/No task [82]. As noted in Section 2, the SAS has been argued to be involved in the consciously controlled inhibition of irrelevant responses in favour of task-appropriate behaviour [19]. Updating is a component of executive function [59,60,83] commonly argued to be highly analogous with working memory. Domain-general updating deficits were found in the sample with dyslexia in this study. In terms of the relevance to the Model of Control of Action, the role of the SAS in working memory has already been considered in Section 4.1. Finally, set shifting was measured using the Plus-Minus task [84], with the group with dyslexia showing a switch cost around 2.5 times greater than the group without dyslexia, indicating a reduced level of cognitive flexibility. This indicates further SAS dysfunction in the group with dyslexia; as discussed in Section 2, the SAS is required for new schema formation when contention scheduling is inadequate (i.e., when the seemingly apposite action is inappropriate). The SAS is called into play to deliberately curb the inappropriate action and provide a suitable one. If the SAS fails at this point, there are errors in behaviour and these usually occur when actions have strong links with specific contexts (e.g., via frequent application) [21]. This can occur even when the action–context link is recently formed [85]. Therefore, in set shifting tasks, where a realignment of recently formed actions to contexts is required, greater time costs likely indicate a delay in the development of a new schema by the SAS.

Further to these laboratory-based measures, the impact of dyslexia on executive function in everyday life was also investigated in the same paper [12]. The self-report Behavior Rating Inventory of Executive Function—Adult Version (BRIEF-A; [86]) was used to compare the relative frequency of day-to-day executive function problems in the same sample of adults with and without dyslexia. Overall, the adults with dyslexia self-reported a higher incidence of everyday executive function problems in the month prior to completing the report. Scores were also compared at a finer-grained level, analysing the different indices and scales making up the BRIEF-A. The adults with dyslexia identified significantly more frequent difficulties on the Metacognition Index, which measured how well working memory is used to solve problems in a planned, systematic, and organised way. No group difference was found on the Behavioral Regulation Index, which assesses the ability to regulate and control one’s emotional responses and behaviour. More frequent problems were reported by the adults with dyslexia on the Working Memory, Plan/Organise, and Task Monitor scales, all three of which load on the Metacognition Index. The Working Memory scale measured how able the respondents felt they were in maintaining information in memory in an active state, thus permitting tasks to be carried out successfully. Links have already been established between working memory and SAS function in Section 4.1 but this result highlights the impact of domain-general working memory problems on everyday life. The Plan/Organise scale measured the participant’s self-reported ability to oversee current and future task demands. The more frequent self-reported problems on this scale in the group with dyslexia could be seen as instances of the SAS’s role in guiding individuals towards their goals and objectives. The Task Monitor scale asked respondents to assess the frequency with which they experienced problems with keeping track of their successes and failures in problem solving and identifying and correcting errors. In this case, there is a clear link to the top-down control of cognition and the putative involvement of the SAS in error correction, as argued by Norman and Shallice [19]. In finding problems restricted to the Metacognition Index alone, the results obtained from adults [12] differed from those taken from children [79], in which more frequent difficulties were reported across both the Metacognition and Behavioral Regulation indices and every scale by parents and teachers of children with dyslexia. This may reflect an age-related difference or the likely greater insights to be gained from self-reports compared with proxy-ratings.

### 4.3. Strategy Selection and Use

Research on the selection and choice of strategies also provides support for an SAS deficit in dyslexia. In one such study, seven- and eight-year-old children who were grouped as either good or poor readers were assessed on a short-term memory task [87]. The children were first shown a series of pictures of objects (e.g., spoon, baby, apple) in a booklet where the experimenter pointed to the pictures in a sequence and the participant was told to remember the sequence. After a delay, the children were asked to point to the pictures in the same sequence. During the delay period, they were observed for evidence of vocal or sub-vocal rehearsal. The task was then repeated but, in this second version, the children were instructed to say the picture names out loud in both the presentation and recall phases. An “inability or lack of inclination” (p. 56) was reported in reading-disabled children to use verbal rehearsal strategies on the non-verbal, pointing-only version of the task. However, the authors found no such group difference in the verbal, point and name condition, a condition which the authors interpreted as facilitating the use of rehearsal. It has also been argued that people with dyslexia may lack the cognitive flexibility to access the metacognitive information which would allow them to recognise the potential usefulness of other potential strategies [88].

There is also evidence from neuroimaging studies for dyslexia-related problems with the self-regulation and management of learning in the absence of support and guidance from teachers [89]. This was indicated by greater functional connectivity found in children with dyslexia compared with typically developing children during a resting state condition. When there was support from an experimenter providing instructions during written composition planning, the children with dyslexia did not differ in functional connectivity from typical reading children.

Group differences in strategy preference have been reported in adults with dyslexia on syllogistic reasoning problems [90]. While there were no group differences found in reasoning accuracy, the participants without dyslexia showed a preference for using verbal strategies in generating their own answers to the syllogisms. Conversely, the participants with dyslexia displayed a preference for spatial strategies. However, it should be noted that more recent evidence suggests that this group-related strategy preference may not necessarily be so fixed. It has been argued that the provision of multiple-choice answer options can help adults with dyslexia to change strategy when reasoning syllogistically [91]. With answer options being provided, the adults with dyslexia solved a higher proportion of problems successfully when they used a mixed verbal and spatial strategy, rather than employing a spatial strategy alone. While it was acknowledged that reasoning strategy choice under conditions requiring self-generated answers was not directly compared with those in which multiple-choice answers were provided, it was proposed that the multiple-choice answers may have allowed the participants with dyslexia to both consider and try out other approaches to problem solving that they may not have been aware of if they had been generating their own answers. The authors suggested that the provision of answer options might support a weaker SAS by providing cues to break out from one strategy to try another approach [91]. This explanation would be consistent with the findings on written comprehension planning considered in the previous paragraph [89].

Such differences or difficulties would seem to extend beyond syllogistic reasoning to performance on working memory tasks [92,93], suggesting that strategy generation or strategy change may be more problematic for people with dyslexia. Strategy deficits have been found in adults with dyslexia when performing a test of spatial working memory [92]. While the adults without dyslexia were able to identify and switch to a new strategy to aid recall under the more taxing conditions requiring the recall of spatial locations in reverse order, the adults with dyslexia did not. However, when they were shown the strategy explicitly and used it as instructed, they performed at a comparable level to the adults without dyslexia. Further evidence of divergent strategy use has been reported in working memory tasks in adults with dyslexia compared with typically developing adults [93]. In this study, recognition ability and serial order retention were assessed in a word-based working memory task. While the control group used verbal maintenance strategies to remember the target words as expected, the group with dyslexia was more likely to use a visual strategy. This maladaptive approach prevents the use of beneficial memory strategies such as chunking [94], creating a greater cognitive load and thus further impairments in attentional abilities such as those involving the SAS [95].

A dyslexia-related strategy deficit has also been proposed [96] in order to explain poorer performance on a design fluency task. Executive fluency tasks, such as the design fluency task, are well-recognised as measures of executive function [63,97,98]. These tasks test the ability to generate items according to defined rules within a specified amount of time. It has been proposed that the reduced output of the children with dyslexia on a design fluency task was due to a reduced effectiveness in the strategies that they could bring to bear on the task [96], with the authors claiming that their task was quite abstract in nature and required creativity when responding to it. They highlighted the task requirement for a response to novel task demands (as noted in Section 2, this is identified as a key aspect of executive function; e.g., [32,33]). Therefore, the need for executive function resources (and, thus, by extension, the involvement of the SAS) seems to be reflected in this argument. Related to this point, an argument has been made for the increased need for a selection mechanism for phonemic fluency performance compared with semantic fluency, due to competition from both habitual use of words based on meaning as well as based on their initial sound or letter [99]. Furthermore, the greater novelty of phonemic fluency tasks over semantic fluency tasks as an explanation (at least partial) of the dissociation in the verbal fluency performance of adults with dyslexia has been identified [100].

The impact of dyslexia on strategy use and preference is not limited to the laboratory setting but is also evident in approaches to studying. For example, learning strategies in university students with and without dyslexia have been studied, with the students with dyslexia obtaining lower scores on a self-report questionnaire measuring awareness and the use of effective learning strategies [101].

Overall, the evidence presented in Section 4.3 supports the notion that people with dyslexia may struggle with initiating the appropriate behaviour required in a novel situation where there is a known goal. This is, therefore, consistent with the concept of deficits in the SAS and the development of new schemata as described in Section 2. Furthermore, these studies have shown that group differences are not evident when the participants are given guidance as to what behaviour would be appropriate for the given situation (e.g., strategy use). This is in line with the description of neuropsychology case studies in which patients with dysexecutive syndrome could improve their performance on tasks requiring novel actions when they were provided with an organised programme of sub-actions with which to approach the task [19]. The breadth of the differences in strategy selection and use presents a challenge to dyslexia theory as it does not sit well with phonological processing deficit accounts (e.g., [3,4,5]). However, there is a clear role for executive function in strategy selection in children [102] and adults [103] in general, and for SAS function in particular [23,31], thus adding to the validity the Model of Control of Action in explaining broader cognitive deficits in dyslexia.

### 4.4. Prospective Memory

Prospective memory is the memory system responsible for retrieving delayed intentions at the point at which they need to be executed [104] or, in other words, “remembering to remember” [105]. Successful prospective memory performance requires two components to function effectively (e.g., [106,107]). The prospective (or planning) component should act at the appropriate moment in time to remind an individual that there is an intention which needs to be acted upon. The retrospective component provides the individual with the detail of what precisely the intention entailed. There are two main types of cue to prompt prospective remembering, namely those that are event-based (where people or objects in the surrounding environment act as reminders that an intention needs to be carried out; e.g., passing a post-box should prompt the intention to post a letter) and those that are time-based (where an intention needs to be carried out at, or by, a particular time in the future; e.g., paying a bill by the end of the week). The use of prospective memory can either be episodic (i.e., for one-off events) or habitual. A role for the frontal lobes in prospective memory has been argued (e.g., [108,109]). More specifically, it has been proposed [110] that different areas of the PFC are involved in different aspects of PM, with lateral PFC being involved in the maintenance of intentions during performance of ongoing tasks, while medial PFC is involved in the detection of prospective memory cues. This evidence led them to propose a “gateway” hypothesis, under which the medial PFC acts to coordinate attention in response to environmental input, while the lateral PFC is called upon for the controlled disengagement of attention from stimuli and its reorientation towards the internal representations of intended actions.

Dyslexia-related prospective memory deficits have been found in both children [111] and adults [112,113,114,115].

In children with dyslexia, more frequent prospective memory problems have been self-reported [111]. More specifically, the group differences were greater when self-cued prospective memory was required rather than when it was environmentally supported. Self-cued prospective memory requires the internal generation of strategies to prompt remembering of the intention at the point at which it is needed. As identified in the literature (e.g., [116]), executive processes are called upon to generate such strategies.

More frequent internally cued prospective memory difficulties have also been identified in the self-reports of adults with dyslexia [114]. Further to this, long-term episodic prospective memory performance, where successful execution of one-off delayed intentions was required over delay intervals of hours or days (such as forgetting to attend scheduled appointments), was also rated as being more problematic by the same adults with dyslexia. In the latter case, a role for the SAS can be identified in setting up schemata to deal with novel task demands. From the self-report ratings, it would appear that this is less effective in adults with dyslexia. In contrast with more frequent problems under these two types of prospective memory demand, the self-reported frequency of problems with habitual prospective memory tasks with short delays between forming an intention and putting it into effect (such as remembering to lock the door when leaving one’s house) did not differ between the adults with and without dyslexia. This pattern would suggest that the more frequent prospective memory problems experienced in daily life by adults with dyslexia are the result of difficulties with adaptive rather than habitual behaviour and highlights one way in which such problems can have a significant impact on everyday life. The respondents with dyslexia also rated themselves as using tools and techniques to support prospective memory more frequently but, despite this increased use, still reported more problems with self-initiated and long-term episodic prospective memory tasks. Ineffective use of such memory aids is suggestive of both metacognitive problems and issues with the use of strategies, as considered in Section 4.3.

In a review of self-report and objective measures of prospective memory, the likely circumstances under which prospective memory is most likely to be affected by dyslexia have been proposed [117]. These are when cues are time-based, when prospective memory tasks are episodic or one-off in nature rather than being repeated or habitual, when the delay between forming an intention and having the opportunity to execute it is prolonged, and when prospective memory performance needs to be self-initiated (through internally generated strategies, such as free recall) rather than being supported by cues in the surrounding environment. With the exception of prolonged delays (which may well also tax executive function more in terms of generating strategic reminders to carry out the task as intended), these characteristics are consistent with the kinds of prospective memory that draw more heavily on executive function [118,119].

A general role for executive function in breaking out from ongoing activity to perform prospective memory tasks has also been identified [120]. More specifically, a cue-monitoring role for the SAS in PM has been proposed [121], with the SAS being called upon to act to permit the individual to break out from ongoing activity upon the appearance of the relevant cue in order to perform the prospective memory task as intended. Under this view, the SAS is required to inhibit the responses to the ongoing task which were controlled by contention scheduling processes [121].

From the study of prospective memory in dyslexia [113,114], it would indeed seem that adults with dyslexia are less likely to remember to break out from ongoing activity to perform a prospective memory task but show no problems in reporting the content of the intention, at least in the shorter-term. This pattern of findings would seem to indicate that the lower prospective memory scores of the group with dyslexia were the result of a failure in the executive ability to interrupt ongoing behaviour (i.e., SAS function [121]), rather than in the retrospective memory ability to retain and recall the task purpose (but see [115], for evidence of prospective memory deficits over a one-week retention period).

## 5. The Neuroanatomical Underpinnings of Executive Function and Their Relation to Dyslexia

With regard to neuroanatomy, there is considerable evidence in favour of a link between executive function and the pre-frontal cortex (PFC). Neurodevelopmental studies have shown that the trajectory of the PFC throughout childhood into early adulthood is linked to an increase in higher-order abilities such as attention, task-switching and inhibition (for reviews, see [43,122]). There is also overwhelming evidence from neuropsychological, neurobiological, neuroimaging and computational studies that control processes related to goal-focused behaviour in novel situations (e.g., selective attention, error monitoring, decision-making, memory, and response inhibition) are related to activation in or damage to the PFC (for reviews, see [73,123]).

There is also considerable evidence for neurobiological atypicalities in children and adults with dyslexia in those brain areas related to executive function (for a review, see [124]). Specifically, dyslexia-related frontal lobe problems have been proposed by several authors [77,125,126].

### 5.1. Executive Dysfunction in Dyslexia: Implications for the Roles of the PFC and the Cerebellum

The Dyslexia Automatization Deficit hypothesis [14] argues that dyslexia-related problems arise from a lack of automaticity and the consequent need to allocate attentional resources consciously. This is in contrast to individuals without dyslexia who would not need to do so. Problems then arise when task demands or circumstances make a greater call on attentional resources. Four general types of skill likely to be prone to disruption in dyslexia have been identified [127]. These are (i) complex skills that require fluency in their component subskills, (ii) time-dependent skills which demand fast processing speed, (iii) multi-modality skills that need different modalities or sources of information to be monitored, and (iv) vigilance tasks which require concentration over time. These increased demands made by each type of skill prevent the individual with dyslexia from using conscious compensation to facilitate performance.

The Cerebellar Deficit Hypothesis was initially proposed by Nicolson, Fawcett and Dean [15] who compared the performance of dyslexic and non-dyslexic children on a time estimation task based on the involvement of the cerebellum in time perception. The children with dyslexia performed more poorly that the children without dyslexia on this task; however, their performance was comparable on a control task (loudness estimation). This was interpreted as early evidence of a cerebellar deficit in developmental dyslexia.

In adults with dyslexia, evidence has been found of reduced activation of the right cerebellar cortex during the learning of novel motor sequences (which took the form of finger movements), as well as the performance of pre-learned sequences [128]. In the same study, evidence of increased frontal activation was also found in the participants with dyslexia, with the authors arguing that this compensated for the reduced activation of the cerebellum. More broadly, the cerebellum has been linked to executive control processes [129] and, thus, there could be some scope for explaining difficulties with executive function within Nicolson et al.’s [15] hypothesis.

Broader deficits in dyslexia are not limited to high-order cognitive abilities but also extends to impairments in eye movements [130], poor balance [15,130,131] and procedural motor task performance [132]. Given the importance of these abilities to procedural learning and the role of the cerebellum in coordinating and fine-tuning these movements, deficits in this brain structure have been proposed to explain the problems apparent in those with dyslexia. As already stated, such atypical behaviours have been interpreted [15] as being the result of impairment in automaticity and a cerebellar deficit hypothesis has been proposed based on a review of behavioural and neuroimaging research that has examined these characteristics in people with dyslexia. It is argued that this hypothesis explains impairments in reading as being due to reduced function related to articulation, visual attention, and eye movements (but see [133]).

This hypothesis has been supported by neuroimaging studies that have shown symmetric cerebellar hemispheres in adults [134,135] and children [136] with dyslexia compared with age-matched controls. In these studies, the degree of differences in cerebellar structure correlated with literacy difficulties, compared with typically developing controls. Similarly, case studies of traumatic brain injury have found that damage to the cerebellum resulted in reading difficulties in both children [137] and adults [138] (but see [139]). Furthermore, adults with dyslexia have shown reduced activation in the cerebellum compared with typically developing controls [140].

Based on the evidence presented here, the cerebellar deficit hypothesis might provide explanations for the literacy and non-literacy impairments observed in people with dyslexia. However, given the focus on the automaticity that the cerebellum permits, it seems pertinent to ask how this might marry with the considerable evidence for cortico-centric deficits in dyslexia (e.g., executive function, strategy use, prospective memory and working memory).

The Model of the Control of Action [19], including the SAS, might go some way to explaining these apparent contradictory cognitive profiles of dyslexia. First, it is necessary to consider the purpose of higher-order cognitive abilities, such as executive function. These higher-order abilities enable, for example, planning, prioritisation and decision making, in order to act in the world in an optimal manner depending on immediate environmental demands [122]. For instance, when we are not able to use automatic behaviour to complete a necessary action, the SAS interrupts automatic behaviour to form new schemata which are then applied to the situation at hand [19]. The executive function framework would argue that we inhibit inappropriate responses, hold and manipulate information in mind, and shift our attention or strategy use to achieve the known goal [64,65]. However, this traditional top-down description of cognitive control considers only declarative knowledge and does not acknowledge our reliance on procedurally learned behaviours to drive all actions [141].

The evolution of the architecture of the brain is based on a need to adapt our actions according to environmental demands [142]. We rely on automaticity for the majority of what we do [143,144]; however, interaction with a changing environment means that we often have to interrupt and control such actions. In this regard, the cerebellum-deficit hypothesis is helpful in explaining dyslexia. The cerebellum enables automatic behaviours by constantly adapting to environmental demands [145]. Repetition of these adaptations increases the precision of the behaviour so that it becomes more resolute and, eventually, automatic. This aligns with the Model of Control of Action and the role of Contention Scheduling in selecting routine actions from an organised network of action schemata (cf. [19]). Any limitations in the ability to create these schemata (e.g., arising from either the slow development or activation of a schema or poorly designed schemata) would result in impaired automaticity. Deficits in the ability to develop and habituate to new schemata might explain difficulty in the development of an automatic skill such as reading (c.f. [14]).

However, when the control of action is first executed by the cerebellum, it must interact with the PFC to anticipate certain outcomes and adapt accordingly to control motor behaviour [146,147]. This interaction is required, for example, to plan, prioritise, and make decisions. If cerebellar function is impaired, then this process might be slower and/or more error prone. This would manifest in deficits in higher-order abilities such as those defining executive function. In this way, the Model of the Control of Action [19] might explain the dual impairments of automaticity and cortical control observed in dyslexia. As described previously, according to this model, contention scheduling and the SAS work interdependently, one informing or disrupting the other in order to achieve a known goal; much like the cerebellum and the PFC interacting to adapt to environmental demands.

This cerebellar–EF interactive account might explain why dyslexia is less prevalent in countries where the language has a shallow orthography (e.g., [148]). When the spelling-sound correspondence is transparent, pronunciation of written word places fewer demands on adaptive processing [149]. Therefore, when instances of spelling-sound incongruency are less frequent in a language, evidence for reading difficulties due to deficits in cerebella-PFC correspondence would be less apparent. However, in English-speaking countries, where orthographies are more opaque [150,151] there is a greater observed prevalence of dyslexia ([148]). This may be due to more frequent instances of incongruent spelling and pronunciation of words (e.g., “ough” or “ead” sounds) (c.f. [151]). This, in turn, might place greater demands on the need for adaptive behaviours governed by the cerebellum and executed by the PFC. If the link between these two brain regions is impaired in some way (e.g., due to the differences in cerebellar structures found in people with dyslexia), then delays or errors in this process would be more evident when reliance on them is more frequent. Given the possible explanations provided for deficits in reading and writing by the interactive model, the links between executive function and these literacy skills are discussed in the next section.

## 6. Reading, Writing, and Executive Function

In the preceding sections, cognitive deficits in people with dyslexia have been reviewed. However, the reason why such impairments might manifest themselves as a difficulty with reading has not yet been discussed in detail. Some explanations have been proposed in a review [152] which sought to better understand the phonological deficits reported in dyslexia. The authors examined a series of studies that investigated various phonological explanations of developmental dyslexia. They interpreted the results as suggesting that there was no impairment in phonological representations. Instead, the authors posited that the deficits are evident in the ability of individuals with dyslexia to access phonological representations under particular conditions, such as when short-term memory storage, time pressure or ‘noisy’ distractors are involved. It is not implausible to consider that each of these conditions might require working memory, fluency and inhibition respectively, so that reduced performance on these tasks might be explained by executive function deficits.

### 6.1. Executive Function and Reading

There is a considerable literature on the role of executive function in the typical development of reading ability. A meta-analysis [153] has examined the relationships between different types of executive function and reading comprehension in children, adolescents and adults. Consistent links were found between reading comprehension and working memory, planning and task-switching and these were stronger in children and adolescents compared with adults. As with much research into executive function, there is variability in the findings. For example, a study with adults found only direct relationships between reading and task-switching but not with other aspects of executive function, such as inhibition and working memory (e.g., [154]). This issue is addressed in more detail in Section 7. Nevertheless, the evidence for a link between executive function and reading ability is substantial but, despite the research highlighting the importance of executive function to this ability, models which attempt to explain reading do not consider this set of cognitive constructs. This omission was highlighted in a review [155], in which a synthesis of existing empirical evidence indicated that abilities such as updating, inhibition and shifting supported reading comprehension. It was concluded that future models should incorporate executive function as an important factor in explaining reading comprehension.

The relationship between executive function and reading ability in people with dyslexia has also been investigated. For instance, the potential role of executive function in reading in dyslexia has been discussed [11], raising in particular the impact of inhibition difficulties on reading. In a different study [156], three executive function tasks were presented to children with and without dyslexia and relationships between performance on these tasks and literacy outcomes were found, particularly for inhibition. However, they found that there was less variance contributed by these executive functions to reading and writing outcomes in children with dyslexia than in children without dyslexia. The authors argued that this might be because the children with dyslexia did not apply their executive function effectively to literacy tasks and would, therefore, benefit from being taught self-regulation strategies and to engage and apply executive functions to reading and writing tasks. This difference in strategy use is in line with the research discussed in Section 4.3 relating to impaired strategy use observed in people with dyslexia and its links to SAS function.

Studies with children have also examined the executive abilities embedded in reading difficulties and found reading-specific inhibition [157] and cognitive flexibility deficits [158] in children who struggle with reading compared with typically developing readers. Varvara et al. [18] highlighted the way in which some executive function tasks (such as spoonerisms) were directly related to the problems with reading shown by children with dyslexia. They linked this relationship to a role for executive attentional control mechanisms in the reading process, arguing for the domain-general nature of this involvement and, more specifically, for SAS dysfunction. Lower inhibition and updating scores have been found to predict the diagnosis of dyslexia in children and also predicted poorer reading ability [159]. However, a predictive relationship for switching was not found in this study.

Given the evidence for the involvement of executive functioning in higher-order writing skills (e.g., [160,161,162]), links to this ability and executive function are discussed in Section 6.2.

### 6.2. Executive Function and Writing

There is some evidence for the link between executive function in children and the development of their expressive and receptive written skills [163]. In typically developing fourth grade children, for example, specific roles for executive function in writing text has been identified [164]. These occur when choosing the appropriate grammatical and lexical representations, organising ideas, inhibiting the production of irrelevant text, updating mental representations of the developing text, and keeping track of the writer’s position in relation to the current sentence and to the text as a whole. In third and fifth graders, a contributory role for executive functions in the development of writing skills has also been reported [165]. When writing, the planning process involves the generation of content and the organisation of ideas and goal-setting [162]. In planning text, information is taken from the task environment and from searches of long-term memory (see, e.g., [166] for an overview of the processes). The translating process involves text generation and transcription. Text generation involves the transforming in working memory of ideas into language representations. These language representations are then converted into written language via transcription processes (including lower-level skills such as spelling and handwriting). The revision process involves the evaluation and alteration of words, sentences or text and involves problem detection and problem correction. More broadly, the ability to organise information and to plan actions to complete a known task are fundamental cognitive skills that make up executive functions in children [167] and adults [168]. With regard to the interactive model of executive function, within which there is a reliance on the development of procedural abilities, the corpus of literature is small. However, there is some evidence that the development of automatic sensorimotor abilities required in writing are reliant on executive control [165].

Lower-order graphomotor, vocabulary and spelling skills have been investigated in dyslexia, in both children (e.g., [169] and adults (e.g., [170,171,172,173]). While finding deficits in spelling and the fluency of handwriting, Connolly et al. found no differences in higher-order writing skills such as organisation and idea generation. However, there are some indications of dyslexia-related executive functioning difficulties in the literature on writing in adults with dyslexia. Both the planning and structuring of essays have been highlighted as problematic for Higher Education students with dyslexia [174]. Indeed, in interviews with university students with dyslexia, 76% reported that they experienced difficulties in organising essays [175]. There are also some objective data in support of these dyslexia-related essay writing problems in adults. More pauses in writing arising from the use of strategies to avoid words that are difficult to spell have been reported in university students with dyslexia [173]. In another study [176], students with and without dyslexia were asked to outline the arguments for and against the legalisation of euthanasia and then to write a newspaper article about the issue. Relative to students without dyslexia, difficulties were shown by the students with dyslexia in building a stable outline of their ideas prior to writing the article and in the effective use of their outlines during the writing of the article itself. As the authors stated, these differences might arise from low-level writing processes relating to spelling and punctuation interfering with higher-level processes, with reduced working memory capacity having a direct effect on planning processes or a combination of the two.

The studies of executive function discussed in Section 6.1 and Section 6.2 provide evidence for a dyslexia-related reduced ability to deal with environmental demands, which in turn interferes with completing task goals, such as those involved in reading and writing. This interpretation aligns with the explanations related to the cerebellum in Section 5.1. Specifically, being able to handle interruptions from the environment, engage strategies to adapt behaviours accordingly, and plan actions all require interactions between the cerebellum and the PFC [146,147].

## 7. Discussion

This paper has drawn together evidence to argue for the role of the Model of Control of Action [19] as a key contributor to the development of reading abilities and broader cognitive deficits related to dyslexia. Impaired bi-directional instruction between the cerebellum and the PFC has been proposed as one potential explanation for dyslexia-related deficits. This might account for documented impairments across a broad range of cognitive domains, from working memory, to strategy selection, to prospective memory, as well as literacy-related difficulties. However, it should be noted that, at present, the evidence is indirect. The next step is to find ways to test SAS function in dyslexia directly and to relate them to measures of reading and spelling ability. Several methods of measuring SAS function have been suggested [177]. However, it should be acknowledged that such direct tests of the SAS are constrained, in the same way as testing executive function is more generally. This limitation is known as the task impurity problem in the measurement of executive function (e.g., [30,64,178]). The task impurity problem highlights the issue that executive function is only identified in the presence of other cognitive processes such as those involved in language, reading or arithmetic as these are usually required to complete executive function tasks. Individual variation in these areas is likely to pollute performance indices which would, ideally, only reflect abilities such as working memory, inhibition and task-switching. The challenge is to identify a way in which to distinguish the interrelated executive processes evoked when undertaking a novel task from the more basic functions required to complete it. This problem is further compounded by the variety of tasks purported to measure executive function, as the same task can be (and, indeed, has been) used to measure different executive abilities, while the same construct has been measured by very different tasks.

These issues contribute to ongoing debates in the literature as to what constitutes executive function. Burgess [30] refers to a lack of process-behaviour correspondence linked to executive function, as it is the coordinator of various cognitive resources, rather than being a single standalone behaviour. Any task is limited in what it can tell us about executive abilities. This is in contrast to the measurement of more clearly defined cognitive processes. For example, there is consistent evidence of the reliability of the Digit Span task and its construct validity with regard to short-term memory (e.g., [179,180] but see [181]). However, executive function is not restricted to a limited set of processes (e.g., hearing digits, letters or words) and a single outcome behaviour (i.e., serial recall of the stimuli). Instead, it represents a complex set of multiple processes that is possibly only limited by the extent to which it is called into play on any given task. This is evident in neuropsychological case studies where an individual can demonstrate typical ability in standard tests of executive function yet exhibit severe executive impairment in their ability to function in the real world [182]. Reliance on executive abilities might, therefore, only be demonstrated by cognitive demands that are yet unknown, such as those one encounters in previously unexperienced situations in the real world. This possibly highlights an issue raised in Section 5.1, that any cortico-centric model of executive function is limited since it does not recognise that we constantly rely on procedural knowledge, even in novel situations. If this were not the case, then any novel element would require us to perform a task as if each constituent component were being conducted for the first time [141,183]. Traditional laboratory-based tasks strive to create near-complete novelty in their design, in order to replicate what are considered to be the demands representative of those faced in the real-world. It might be the case that measures of executive function must evoke automatic behaviours that are affected by, but not completely changed by, external demands.

This review has presented the evidence for deficits in both automaticity and cortical control, attempting to explain them through the lens of the interactive model of executive function. It has proposed the Model of Control of Action [19] as a framework for further investigation of this proposition. This is required, as the research on executive function reviewed in this paper highlights the general absence of a theoretically driven approach in which exploring executive function deficits in dyslexia are considered explicitly. Executive function in dyslexia has tended to be explored individually or in a “patchwork” fashion which consequently lacks theoretical coherence [12]. This observation is not meant to detract from the important work that has been done to identify areas of impairment in dyslexia but, to move the area forward, a more theoretically informed approach is needed in order to address this aspect of dyslexia.

Although the framework of Miyake and colleagues’ [64,65,83] has provided a useful means for exploring executive function in a more theoretically driven way, it does not appear to take understanding further than indicating the likely breadth of executive function problems in dyslexia. The interactive model has the potential to do exactly this. By investigating the observed automaticity and higher-order deficits in dyslexia and the interdependent links between the cerebellum and the PFC, this approach might go some way to explain, not just the deficits themselves, but also how the development of skills such as reading and writing might rely on interactive and adaptive behaviour with regard to the surrounding environment.

The interactive model aligns to some degree with the Problem-Solving Framework [184]. This framework was developed to understand the functionally distinct aspects, and temporal phases of, executive function when problem-solving (i.e., problem representation, planning, execution, and evaluation). From this perspective, problem-solving is viewed as being the ultimate goal of the individual. While the framework was formulated to explain the development of executive function in preschool children, it may, nevertheless, provide a useful means by which to identify where particular dyslexia-related executive function difficulties lie from a functional rather than a descriptive perspective (cf. [64,65,83]) and where, specifically, these problems may lie temporally in the time-course of problem-solving.

## 8. Conclusions

The evidence presented in this paper provides a reasonable argument for theories of dyslexia to consider an interaction between procedural learned behaviour and higher-order abilities, inherent in the Model of Control of Action [19] and their subsequent impact on reading and spelling abilities. Connected to this, there is the evident need for improved measures of executive function designed to tap this interaction. If impairment in this interaction can be identified in the early years of childhood, it might enable prompt identification of future difficulties with literacy, thus allowing timely interventions and reasonable adjustments to be put into place. Further to this, there is value in considering how findings from this paper could aid the understanding of the cognitive profiles of other neurodevelopmental conditions in which cerebellar and PFC deficits have been identified, such as attention deficit hyperactivity disorder and autism spectrum disorder [185].

As well as indicating the need for a fuller consideration of executive function deficits in dyslexia theory, the current paper has also highlighted the everyday impact of dyslexia-related executive function problems on everyday life across different cognitive domains. Difficulties in these areas are likely to have a broader impact on the life chances of individuals with dyslexia. This is because executive function problems can have a negative impact on success in both educational contexts [153,155,162] and in employment [186,187]. The issues raised in this paper, thus, have significant implications, both theoretical and applied.

## Data Availability

Not applicable.

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
