# Peer review of "Automaticity and Executive Abilities in Developmental Dyslexia: A Theoretical Review"

_brainsci, 2022, doi:10.3390/brainsci12040446_

Round 1
Reviewer 1 Report
Review for brain science
This review examined evidence from studies of working memory, executive function, strategy selection, prospective memory, and everyday cognition and summarized evidence from the theoretical perspective of the Model of the Control of Action. This is an important topic. However, it is hard to gauge the contribution of the current theoretical review. The manuscript needs a major revision before it can be accepted for publication.
Abstract:
- Please state the type of this theoretical review (narrative review, or descriptive review, or scoping review). Refer to Table 2 in http://dx.doi.org/10.1016/j.im.2014.08.008 (no explicit study selection – should be narrative review). For descriptive and scoping review types, the authors need to add
- Please be specific about the contribution of this theoretical review to the current literature. What are the main points summarized from the current literature?
Introduction:
- First sentence: writing and spelling might be related to reading difficulties. “Developmental dyslexia (DD) is usually characterized by difficulties with accurate and/or fluent word recognition and by poor spelling and decoding abilities.” https://www.idaontario.com/about-dyslexia/ Suggesting: remove writing in this sentence. If writing is included, it is literacy. Literacy difficulties include both reading and writing. DD usually focuses on reading abilities.
- Why is this theoretical review needed? Please briefly highlight this point in the first paragraph of introduction. What is the intention of using SAS as the guiding framework for this review? The rationale for this review is unclear. The authors stated the outlines of the review. But it seems to be disconnected why they choose to present their summary in this order and organization.
The Model of the Control of Action
- A figure of this model should be added for clarity.
- SAS comes from neuroimaging – this claim lacks detailed support. The Stroop task and stop-signal task are both targeted for inhibition control. Be explicit about the connection to SAS theory.
SAS dysfunction in dyslexia
- Developmental Dyslexia is in the title. But this section is only focused on dyslexia in adults. Please add pediatric population. Or making the title broader.
- How are the presenting sequences chosen? Unclear in the current version (WM, EF, Selection, Prospective memory, everyday cognition).
Further evidence – Working memory
- The figure mentioned earlier can include the domains included in this review. It is unclear why working memory, executive function, etc. are included. In addition, for some readers, working memory can be included into executive function (refer to Miyake 2012: DOI: 10.1177/0963721411429458 and https://doi.org/10.1080/00461520.2017.1309295)
- The first paragraph is out of place. It focuses on multicomponent model not working memory.
- Literature review on working memory is not comprehensive at all. The connection between working memory and SAS is not clearly described. It seems to be indirect connection.
Further evidence – Executive function
- SAS is analogous with executive function. The authors intertwined working memory in this section too. It is confusing for readers.
Further evidence – Strategy selection and use
- Without clearly stating SAS and its relationship with other cognition mentioned in the beginning of the manuscript, it is hard to follow the logic presented in the current version of manuscript. The sections seem to be stitched together without comprehension thoughts. The review of literature sometime is like a laundry list and lacks connections.
Further evidence – Prospective memory
- There is little mention of SAS in the last paragraph of this section. Please state in the first paragraph and then expand more.
Further evidence –Everyday cognition
- It is less developed. Not sure why this section is necessary.
The neuroanatomical underpinnings of executive function and their relation to dyslexia.
- 1. Executive function, dyslexia, and the cerebellum
This title needs to be changed to “5.1. Executive function, dyslexia, and PFC & cerebellum”
- A figure can be helpful for this section too.
- Reading, writing, and executive function
- There is no smooth transition from one section to the other. It seems to be a sudden topic change. Suggest removing writing portion. Or develop the writing portion more in depth.
Discussion
- The first sentence is super long. Please cut to short sentences. (Break down)
- The ongoing debates in the literature as what constitutes executive function is valid. The second paragraph needs to be expanded to highlights contributions from the current manuscript.
- Discussion about Miayke’s work and Problem-solving framework is superficial. Please elaborate what the current manuscript is different from the two theories.
Conclusions
- “The issues raised in this paper, thus, have significant 755 implications, both theoretical and applied.” Can the authors be specific? What do they mean?
Reviewer 2 Report
This is a nicely written and comprehensive review that acknowledges a nice amount of prior work on the mechanisms and deficits in dyslexia. Especially as the field turns its attention more to executive function and working memory issues, a review like this should be useful.
My main criticism is that there seems to be no mention or discussion of the growing acceptance of heterogeneity in this population. The authors even slip into the traditional approach to defining dyslexia in the introduction by discussing it as a disorder of phonological processing, when this is not the only deficit (see the double deficit model work). It is noble and traditional to search for a single unifying theory of dyslexia, but I think we need to be mindful of the evidence supporting multiple pathways to this diagnosis. Such a view requires that multiple theories be correct, with one theory explaining a different subgroup. If the authors believe their theory accounts for all subgroups (ex. phonological awareness deficits AND rapid automatized naming), a section explaining their perspective on this would be beneficial.
The authors also spend relatively little time describing SAS and how the prior work on EF function may or may not speak directly to SAS deficits. Since SAS seems to be the foundational element to their theory, integrating this element throughout the review would be helpful.
Round 2
Reviewer 1 Report
Thank you for the authors' efforts in revising the manuscript.